



# Aquaplanet simulations with winter and summer hemispheres: The setup and circulation response to warming

Sebastian Schemm[1] and Matthias Röthlisberger[1]

[1]Institute for Atmospheric and Climate Science, ETH Zürich, Zurich, Switzerland

**Correspondence:** Sebastian Schemm (sebastian.schemm@env.ethz.ch)

**Abstract.** To support further understanding of circulation changes in a warming climate, an idealised aquaplanet model setup containing summer and winter hemispheres is presented and the results of circulation changes under warming are discussed. First, a setup is introduced that allows aquaplanet simulations with a warmer and a colder hemisphere, with realistic looking summer and winter jet streams, storm tracks and precipitation patterns that are similar as in observations, with a more intense

and more equatorward storm track in the winter compared to the summer hemisphere. The sea surface temperature (SST) distribution used is inspired by the June-July-August zonal mean SST found in reanalysis data, and is flexible to allow control of the occurrence of a single or double Inter-Tropical Convergence Zone (ITCZ). The setup is then used to investigate circulation changes under uniform warming, motivated by recently discussed research questions. First, we show that jet stream waviness decreases under warming when compared on isentropes with maximum wind speed or on isentropes at similar height in

pressure space. Jet stream waviness increases under warming when compared at similar-valued isentropes, but solely because the corresponding isentrope is closer to the surface in the warmer climate and waviness increases downward in the atmosphere. However, we also observe a waviness increase at isentropes at very high levels (e.g., 350 K) in the colder hemisphere, which does not appear to be due to a change in height. A detailed analysis of the changes in wave amplitude for different wave numbers confirms that the amplitude of large waves increases with warming, while that of short waves decreases. The reduction in wave

amplitude of short synoptic waves is found to dominate in the jet core region, where jet waviness also decreases, and is more pronounced on the equatorward side of the jet. Long waves increase in amplitude on the poleward side of the jet and at upper stratospheric levels, consistent with increased jet waviness at these levels. The projected increased amplitude of planetary waves and the reduced amplitude of synoptic waves are thus clearly apparent in our aquaplanet simulations and thus do not require zonal asymmetries or regional warming patterns. During so-called high-amplitude wave events, there is no evidence

for a preferential phase of Rossby waves of wavenumbers five or seven, indicating the crucial role of stationary waves forced by orography or land-sea contrast in setting previously reported occurrences of preferential phases. Finally, we confirm that feature-based block detection requires significant tuning to the warmer climate to avoid the occurrence of spurious trends. After adjustment for changes in tropopause height, the block detection used here shows no trend in the summer hemisphere and an increase in blocking in the colder hemisphere. We also confirm previous findings that the number of surface cyclones tends to

decrease globally under warming and the cyclone lifetimes become shorter, except for very long-lived cyclones.



## 1 Introduction

The influence of anthropogenic climate change on the atmospheric circulation and associated weather extremes has received considerable attention. Preliminary evidence is presented in the literature that the Arctic amplification (AA) reduced the meridional temperature gradient (Chemke and Polvani, 2020), weekend the westerlies (Coumou et al., 2015) and the Rossby wave amplitude (Fragkoulidis, 2022). Typically, the observed trends, for example in temperature extremes, are strongest during boreal summer (Christidis et al., 2014; Francis and Vavrus, 2015; Coumou et al., 2015). Particularly debated research questions concerning changes in the summer circulation are changes in the waviness of the jet stream (e.g., Francis and Vavrus, 2012, 2015), changes in the persistence and frequency of, for example, blocked weather situations (e.g., Woollings et al., 2018), and related to this the preference for specific phase positions of certain wavenumber waves during high-amplitude wave events (e.g., Kornhuber et al., 2020), the amplification of waves with small wave numbers (e.g., Chemke and Polvani, 2020), and changes in the intensity, number and tracks of surface cyclones (e.g., Priestley and Catto, 2022).

Regarding the jet waviness, it was first suggested that weaker westerlies allow for amplified meanderings of the jet stream causing an increase in hot and cold temperature events (Francis and Vavrus, 2012, 2015) as well as more persistent and blocked weather (Coumou et al., 2015; Francis et al., 2018). Changes in jet waviness and its causal link with regional patterns of warming such as the AA, or even just uniform warming, are however still "up in the air" (Francis, 2017; Cohen et al., 2019). In particular because some trends eventually result solely from incorrectly adjust diagnostics to the warmer climate (Barnes, 2013). Indeed, idealized zonally symmetric dry simulations of polar warming (i.e., reduced meridional temperature gradients at the surface) identified a reduction in blocking frequencies and wave amplitudes and not an increase (Hassanzadeh et al., 2014). Martin (2021), however, analysed in three reanalysis products the jet waviness on the height of the jet maximum, rather than on a constant vertical height level, and identified an increase since the 1960s. Very recently, Moon et al. (2022) proposed a mechanism for an increase in waviness of planetary waves (wave numbers 1 to 4). In their linearized model of balanced planetary wave dynamics, which neglects synoptic baroclinic growth and prescribes a zonal asymmetric surface heat source, the planetary wave amplitude indeed increases when the vertical wind shear decreases. A further proposed theoretical underpinning of the potential trends towards more waviness, blocked and persistent weather due to wave resonance (Petoukhov et al., 2016) does not seem to withstand scrutiny of the assumptions underlying the theory (Wirth, 2020; Wirth and Polster, 2021). Although wave resonance may well occur in the atmosphere if the wavenumber of free tropospheric Rossby waves approaches the stationary wavenumber forced by topography (Charney and Eliassen, 1949), persistence of high-amplitude weather is often associated with locally recurrent Rossby wave patterns (Röthlisberger et al., 2019) rather than circumpolar wave patterns as is required by resonance (Petoukhov et al., 2016) or as found in planetary wave models (Moon et al., 2022). At the same time however, there is evidence that large waves become less stable while small waves become more stable under warming (Chemke and Ming, 2020) and the former might indicate a larger contribution to the overall jet waviness from planetary waves. The underlying mechanism of the "large-get-stronger, small-get-weaker" wave response to warming likely relates to the different vertical extend of large and small-scales waves and varying trends in baroclinicity at different vertical levels in the atmosphere (Rivière, 2011). Indeed, Martin (2021) suggested that the upward trend in waviness identified in three





reanalysis products may not result from Arctic amplification (and reduced vertical wind shear) but from an increase in the frequency of anticyclonically tilted waves at the jet stream level that push the jet poleward and have larger length scales than cyclonically breaking waves (Rivière, 2011).

The situation is complicated because regional trends may oppose each other, for example, Francis et al. (2018) identified an increase in persistence of weather regimes over North America, while Huguenin et al. (2020) found no trend in persistence

in circulation regimes over Central Europe (neither in summer nor in winter). Consistent with that, globally, the trend in the frequency of blocking is not yet conclusive (Woollings et al., 2018). Regional persistence trends over Europe can be reconciled with the projected poleward shift and eastward extension trend of the North Atlantic (Kyselý and Domonkos, 2006; Hoskins and Woollings, 2015; Harvey et al., 2020) and thus can be considered only as "passive" changes and not as a global reduction of blocks. From this projected storm track shift, one can also expect a regional reduction in the number of surface cyclones over

Europe. However, the number of surface cyclones appear to reduce even globally during summer (Zappa et al., 2013; Chang et al., 2016) and winter (Priestley and Catto, 2022). For surface cyclones there is some evidence from previous wintertime idealized warming experiments with atmospheric general circulation models (AGCM) that the number of cyclones decreases globally even under simple warming (Sinclair et al., 2020; Schemm et al., 2022) and the reduction in cyclone numbers hence appears the most robust of the above discussed changes.

As expected with these topics, the variety of methods to detect blocks, cyclones, waviness, weather regimes or persistence is so large that the trends are not always clear cut (Kučerová et al., 2016). Even more so, studies have for example questioned whether weather regimes – such as blocks – defined by statistical cluster techniques are regimes that are per see more persistent in the sense of slowly evolving states of the atmospheric circulation within a phase space (Fereday, 2017). Uncertainties in cyclone track statistics arise from various cyclone tracking methods (Neu et al., 2013). Uncertainties in trends of jet waviness

may result from the fuzzy concept of waviness in general and the application of different method to different height levels (Barnes, 2013). For example, Martin (2021) employs a geometric definition of jet waviness and applies it to the height of the maximum wind speed. Hence the height level changes over time. The finding is an upward trend in waviness since the 1960s. Blackport and Screen (2020) use a related method but report no clear upward trend. Eventually the difference is because their study is based on a different height level (i.e, 500 hPa), which is held constant over time but under warming changes its height

above the surface. The exact caused of the often contrasting results remains unclear.

The picture that emerges from the above overview is that idealized studies of summertime and wintertime circulation changes are potentially a useful stepping stone of intermediate complexity to (a) establish a baseline response of the summertime circulation to simple warming, (b) to explore the need to adapt feature-based detection methods to a warmer climate, and (c) to make the data available for the community to explore the performance of various further diagnostics. To this end we propose

an idealised aquaplanet setup that comprises a warmer summer-like and colder winter-like hemisphere shaped exclusively by the underlying SST distribution. In the warmer hemisphere, the equator-to-pole SST gradient is weaker than in the colder hemisphere, favouring the development of weaker summer-like jet streams and storm tracks that are similar to observations located further poleward. Furthermore, similar to the boreal summer observations, the maximum SST values are shifted into the warmer hemisphere, and the cross-equatorial SST gradient can be used to control the occurrence of a single or double



ITCZ (Bischoff and Schneider, 2016; Adam et al., 2016). Such a hemispheric asymmetric SST distribution, which can be used for the simulation of a summertime aquaplanet, is potentially also useful for a variety of further research questions related to circulation changes in a warmer climate or serve as a testcase for numerical model development. As far as we know, no summertime aquaplanet experiments have been systematically studied so far.

      This study therefore consists of two main parts: In the first part we introduce the setup, design and mean climatologies of
an aquaplanet simulation with winter and summer hemisphere. In the second part we demonstrate the usefulness of our APE setup by going through the above discussed summer circulation changes and by testing their existence in our APE simulations. Specifically we address the following questions:

   – How is the jet waviness changing under simple uniform warming and what is the role of changes in the height of the investigate isentropic level (e.g., Barnes, 2013; Francis and Vavrus, 2015)?

– Is a "large-get-stronger, small-get-weaker" wave response to warming (e.g., Chemke and Ming, 2020) apparent in APE simulations and at which altitude do the changes occur?

   – Is there a preferred phasing of high-amplitude large-scale waves even in the absence of zonal asymmetries in the mean state (Kornhuber et al., 2020)?

   – How is the global frequency of atmospheric blocking affected by uniform warming (e.g., Woollings et al., 2018)?

– How is the frequency of surface cyclones changing globally in a simple warming experiment (e.g., Priestley and Catto, 2022)?

      It goes without saying that one cannot expect a summertime-like APE to reproduce all observed and simulated summer circulation changes found in fully-coupled Earth system models, which include an interactive ocean and topography. Rather, the presence of some changes and the absence of others may indicate the relative importance of components that were not
considered in the simulation for the above circulation changes. Here, we thus use an APE setup for constraining the palette of possible causes of circulation changes, e.g., for ruling out or emphasising the importance of changes in zonal asymmetries such as land-ocean contrasts, the AA, changes in the ocean circulation or topography as causes for simulated circulation changes.

## 2 Diagnostics

### 2.1 Jet waviness

In order to quantify the waviness of the jet stream, we opt for the geometric definition of waviness as in Röthlisberger et al. (2016) and Martin (2021). The jet stream axis is defined as the 2 PVU contour on an isentropic surface. The absolute value of the latitude difference between adjacent points along the contour is summed. This geometric definition of waviness follows the proposal of Röthlisberger et al. (2016), who related this type of waviness measure to the occurrence of extreme surface weather where the type of extreme (wind, precipitation, or temperature) associated with high waviness is shown to vary regionally. A





zero waviness means a pure zonal flow and there is no maximum in the definition of this waviness similar to the method of Martin (2021). At times, several closed 2 PVU contours are detected, for example, in the presence of additional PV cutoffs. In this case, the algorithm selects the longest contour for computing the waviness. Following suggestions from the previous literature (Barnes, 2013; Martin, 2021), we quantify jet waviness changes in three different ways: (1) we compare jet waviness on the same isentrope between a control and a warmed simulation, (2) we compare jet waviness on two different isentropes, which, however, in the zonal mean both are at the same height in pressure space, and (3) for each time step and in both simulations we choose the isentrope for which the maximum wind speed is found.

## 2.2 Rossby wave amplitudes

To further examine changes in Rossby wave dynamics under uniform warming, we also perform analyses inspired by Chemke and Ming (2020). For all simulation setups (see below) we compute the Fourier transform of the meridional wind ($v$) over longitude for each 6-hourly time step $t$ at each pressure level $p$ and latitude $\phi$. Then we compute the amplitude $A_k(t,p,\phi)$ of the waves for each wavenumber $k$ as the modulus of the respective Fourier coefficient, i.e., $A_k(t,p,\phi) = |\nu(k,t,p,\phi)|$, whereby $\nu(k,t,p,\phi)$ is the respective coefficient for wavenumber $k$. Finally, we compare the wave amplitudes between the warmed and control simulations by computing the amplitude difference

$$\Delta A_k(p,\phi) = \overline{A_k^{warming}(t,p,\phi) - A_k^{control}(t,p,\phi)}, \tag{1}$$

where the overline denotes the time mean. Due to the rise in tropopause height as well as vertically differing changes in baroclinicity expected under warming (e.g., Rivière, 2011; Barnes, 2013), we are particularly interested in the vertical structure of the Rossby wave amplitude changes.

## 2.3 Rossby wave phases

To examine the existence of a preferred phasing of high-amplitude waves we repeat and slightly expand the analyses of Kornhuber et al. (2020) for our simulations. We first compute Hovmöller diagrams of the 7-day averaged 300 hPa meridional wind, averaged between 35 and 65° latitude of the respective hemisphere (hereafter denoted as $v_{hov}(\lambda,t)$, whereby $\lambda$ is longitude). A Fourier transform over longitude is then also applied to $v_{hov}(\lambda,t)$ and the resulting (complex) Fourier coefficients for wavenumbers 4–8 ($\nu_{hov}(k,t)$) are used to examine a preferred phasing of high-amplitude waves. Specifically, we again first compute the amplitude of each of these wavenumber waves for each week as $A_{hov,k}(t) = |\nu_{hov}(k,t)|$. Then, the phase $\Phi_{hov,k}(t)$ (in radians from $-\pi$ to $\pi$) is computed as

$$\Phi_{hov,k}(t) = \arctan2\left(\text{Re}[\nu_{hov}(k,t)], \text{Img}[\nu_{hov}(k,t)]\right). \tag{2}$$

As in Kornhuber et al. (2020) we then compute for wavenumbers 5–8 the standard deviation $\sigma_{hov,k}$ of $\Phi_{hov,k}(t)$ and define weeks with "high-amplitude waves" as weeks for which $\Phi_{hov,k}(t) > 1.5 \cdot \sigma_{hov,k}$. For the roughly forty seasons per simulation setup (see below) this yields between 36 and 47 weeks with high-amplitude waves per wavenumber (compared to 44–47





in Kornhuber et al. (2020) for an equally long data record). The number of weeks with high-amplitude waves in a specific wavenumber is hereafter referred to as $N_k$.

To examine whether there is a statistically significant preference for particular phases during the high-amplitude weeks of any of the wavenumbers, we extend the analysis of Kornhuber et al. (2020) with a bootstrapping test. We first compute a histogram of the $\Phi_{hov,k}(t)$ values for each wavenumber with eight bins of width $\pi/4$. Then, we randomly select 10'000 times

$N_k$ weeks. For these sets of $N_k$ weeks we again compute histograms of the respective $\Phi_{hov,k}(t)$ values. Here we identify a statistically significant preferred phasing when the count of the actual high-amplitude $\Phi_{hov,k}(t)$ values in one bin of the histogram lies outside the 2.5 to 97.5 percentile of the respective counts of the randomly generated histograms.

## 2.4 Blocks

The identification of atmospheric blocks is based on persistent, vertically averaged, upper-tropospheric potential vorticity

(VAPV) anomalies (Schwierz et al., 2004; Croci-Maspoli et al., 2007; Sprenger et al., 2017). Specifically, a northern (southern) hemispheric block is identified as a coherent area with negative VAPV anomaly of less than -1.0 PVU ($1\,\mathrm{PVU} = 10^{-6}\,\mathrm{K\,kg^{-1}\,m^2\,s^{-1}}$) and persisting for at least 5 days. Note that the commonly used threshold of -1.3 PVU (e.g., Croci-Maspoli et al., 2007) is not suitable for blocking identification in summer. To allow for the detection of summertime blocks, the less restrictive threshold of -1.0 PVU is used here, as in Röthlisberger and Martius (2019). Likewise in the Southern Hemisphere, where potential vorticity

is generally negative, a VAPV anomaly is required to be in excess of 1 PVU.

In the basic configuration of the blocking identification, vertical averages are computed for the layer between 500 hPa and 150 hPa, hereafter referred to as $\mathrm{VAPV_0}$. Since changes in the tropopause height, as evident in the warmed simulations (see Fig. 4), can profoundly impact the climatology of VAPV and by that the strength of VAPV anomalies (Croci-Maspoli et al., 2007), the blocking identification in the warmed simulations is additionally performed with vertically shifted layers.

Three additional layers are considered: layers shifted by 25 hPa and 50 hPa upward, $\mathrm{VAPV_{25}}$ (475 hPa - 125 hPa) and $\mathrm{VAPV_{50}}$ (450 hPa - 100 hPa), respectively, and an adjusted $\mathrm{VAPV_A}$ defined as the mean of $\mathrm{VAPV_0}$ and $\mathrm{VAPV_{25}}$.

Model output of potential vorticity is available on pressure levels in intervals of 25 hPa. Climatological and hemispheric (poleward of 30°N or 30°S) averages of $\mathrm{VAPV_{25}}$ and $\mathrm{VAPV_{50}}$ in the warmed simulations do not closely match $\mathrm{VAPV_0}$ in the control simulation. In contrast, the differences between climatological and hemispheric averages of $\mathrm{VAPV_A}$ in the warmed

simulations and $\mathrm{VAPV_0}$ in the control simulations are reasonably small.

## 2.5 Cyclones

The diagnostics introduced above are complemented by feature-based identification of cyclones. Surface cyclone detection is based on the sea-level pressure contour search approach by Wernli and Schwierz (2006) and refined for merging and splitting cases following Sprenger et al. (2017). Minima in sea-level pressure are accepted as cyclones if enclosed by a SLP contour and

traceable for at least 24 hours in 6-hourly data. Deepening rates are extracted based on 6-hourly SLP changes along a track and genesis and lysis are defined as the first and last time step along a track, respectively. The total number of cyclone tracks is about 35'000 tracks in the control simulations and 32'000 in the warmed simulations. Most cyclone detection scheme agree





on the tracks and existence of deep systems, while there is larger uncertainty related to the exact location of cyclogensis, lysis,
and tracks for shallow and short-lived systems (Neu et al., 2013; Walker et al., 2020; Roebber et al., 2023). For the purpose of
our study, the exact track and genesis or lysis locations are not relevant and the changes in number of tracks are qualitatively
compared to previous studies using different detection methods (Sinclair et al., 2020).

All five of the diagnostics introduced above have been used in previous literature and are applied here to data interpolated to
a 1° by 1° grid with a 6 hour temporal resolution.

## 3   An aquaplanet with a cold and a warm hemisphere

APEs are numerical modelling simulations in which the entire surface of the Earth is covered by water, which sometimes
features additional zonal asymmetries. Examples include warm or cold sea surface temperature (SST) anomalies (Brayshaw
et al., 2008; Sampe et al., 2013; Schemm et al., 2022) and SST gradient changes (Brayshaw et al., 2011a; Williamson et al.,
2013), or idealized topography and continents (Brayshaw et al., 2011a). Including a zonal surface asymmetry into an APE
allows to reproduce the triple pattern of change seen in CMIP future projections of the North Atlantic storm track, which
does not occur in zonally symmetric APEs (Schemm et al., 2022). Further, APEs are frequently used to test new numerical
schemes or model grids and established themselves as a standard test for model inter-comparison studies (Blackburn et al.,
2013), including studies focusing on tropical dynamics (Nakajima et al., 2013). Most APE simulations are inspired by the
boreal wintertime climate. Below, we propose a setup that has a summer and a winter hemisphere at least in its SSTs.

### 3.1   The generalised "Qobs" SST distribution

At the heart of an aquaplanet simulation is the SST distribution. In the absence of an interactive ocean and land surface, the SST
distribution shapes the global asymmetries in near-surface temperature and thus the location of the major baroclinic zones, the
strength of the jet stream, and the position and shape of the ITCZ. Most APE studies rely on the symmetric distribution of sea
surface temperature (SST) originally proposed by Neale and Hoskins (2000) and known as "Qobs", or the slightly extended
version of it as defined in Brayshaw et al. (2011b) and known as "Qobswide". The design of both follows the zonal mean
SST of the Northern Hemisphere (NH) from December to February (DJF). The zonal mean SST of the NH is mirrored at
the Southern Hemisphere (SH) to obtain a symmetric SST distribution with a peak, where the peak is located at the equator.
Radiation is typically set to the solar equinox. In this way, both hemispheres are in their initial conditions perfectly symmetric
with respect to the radiation and SST distributions and can be considered as one long time series, NH winter-like, which is an
attractive choice from a computational point of view to obtain a perpetual winter.

The SST distribution in this study is based on a generalised form of the established "Qobs" distribution of Neale and Hoskins
(2000), which allows to mimic a summer and winter hemisphere. We define the generalised "Qobs" SST distribution as follows,

$$T_{sst}(\phi) \quad = T_{\text{melt}} + T_{\max}\left[w_1 f_2(\phi) + (1 - w_1) f_4(\phi)\right] \tag{3}$$



where $w_1$ is a weighting parameter, $T_{\mathrm{melt}}$ the freezing temperature of water, $T_{\mathrm{max}}$ the maximum SST and the latitudinal variation $f(\phi)$ is modelled in each hemisphere by,

$$f_2(\phi) \quad = 1 - \sin^2 \left( 90° \left[ \tfrac{\phi - \phi_{\mathrm{max}}}{\phi_0 - \phi_{\mathrm{max}}} \right] \right), \tag{4}$$

$$f_4(\phi) \quad = 1 - \sin^4 \left( 90° \left[ \tfrac{\phi - \phi_{\mathrm{max}}}{\phi_0 - \phi_{\mathrm{max}}} \right] \right), \tag{5}$$

where $\phi_0$ is the latitude where the SST distribution becomes in the respective temperature $T_{\mathrm{melt}}$ and $\phi_{max}$ is the latitude where the SST distribution reaches its maximum $T_{\mathrm{max}}$ (here set to $28°$). The transition between a single and double ITCZ is controlled by the weighting parameter $w_1$. The occurrence of the transition for a given weighting parameter is highly model dependent and the proposed SST distribution can thus be used in future inter-model comparison studies. The parameters can be used to fit the SST distribution to, for example, zonal mean reanalysis data.

We use two weighting parameters to probe the sensitivity of the results to the shape of the SST. First, we fit the generalised "Qobs" distribution to the JJA zonal mean SST distribution obtained from ERA5 using $w_1 = 0.5$, $\phi_{max} = 5°$ and $\phi_0 = 67°$ in the colder and $\phi_0 = 85°$ in the warmer hemisphere (Fig. 1a). Second, we choose a steeper SST distribution ($w_1 = 1$) that also peaks in the warmer hemisphere at $5°N$. Because the observed SST distribution during JJA approaches $T_{\mathrm{melt}}$ at lower latitudes in the winter hemisphere, we set $\phi_0 = 60°$ in the colder and $\phi_0 = 90°$ in the warmer hemisphere (Fig. 1b). As shown below, the first SST distribution results in the formation of a doule ITCZ, while the later in a single ITCZ.

### 3.2 Model setup and mean climate

*ICON model and initial conditions*: The idealized experiments in this study are performed with the non-hydrostatic weather and climate prediction model ICON (Zängl et al., 2014) in its version 2.6.5. The equivalent grid spacing of the icosahedrical grid is approximately $80\,\mathrm{km}$ and the time step is $360$ seconds. The model top is at $65\,\mathrm{km}$ and spanned by $70$ vertical levels using the SLEVE coordinate system (Schär et al., 2002). The vertical layer thickness is $20$ meters near the surface and increases to approximately $200$ meter at $1.5$ kilometre altitude, $500$ meter at $8\,\mathrm{km}$ altitude, $1000$ meter at $20\,\mathrm{km}$ altitude and $3000$ meter at $60\,\mathrm{km}$ altitude. The subgrid-scale parameterizations include the ecRad radiation scheme developed at ECMWF (Hogand and Bozzo, 2018) and radiation is computed on a horizontal grid that is reduced in space by a factor of two and is updated at every second time step. Further, a one-moment two-category microphysics scheme (Doms et al., 2011), a deep convection parameterization Tiedtke (1989) and a turbulent transfer sub-gridscale model based on (Raschendorfer, 2001). Finally, non-orographic gravity wave drag is parameterized following Orr et al. (2010). This set of subgrid-scale parameterization and initial conditions is similar as in Schemm et al. (2022). The model is initialized with temperature, pressure, and wind according to Jablonowski and Williamson (2006) and simulates $10$ years, resulting in $40$ seasons with summer-like SSTs in the NH and $40$ seasons with winter-like SSTs in the SH.

*Radiation*: In a setup with fixed SSTs and no land surface, radiation has little effect on the climatological mean surface temperatures and, as in most aquaplanet experiments, the solar zenith angle is set to perpetual equinox. Both hemispheres thus receive the same radiation. The climatological differences between the two hemispheres originate thus from the SST distribution alone. Indeed, when we repeat the simulations with the radiation fixed to the summer solstice in the NH the effect





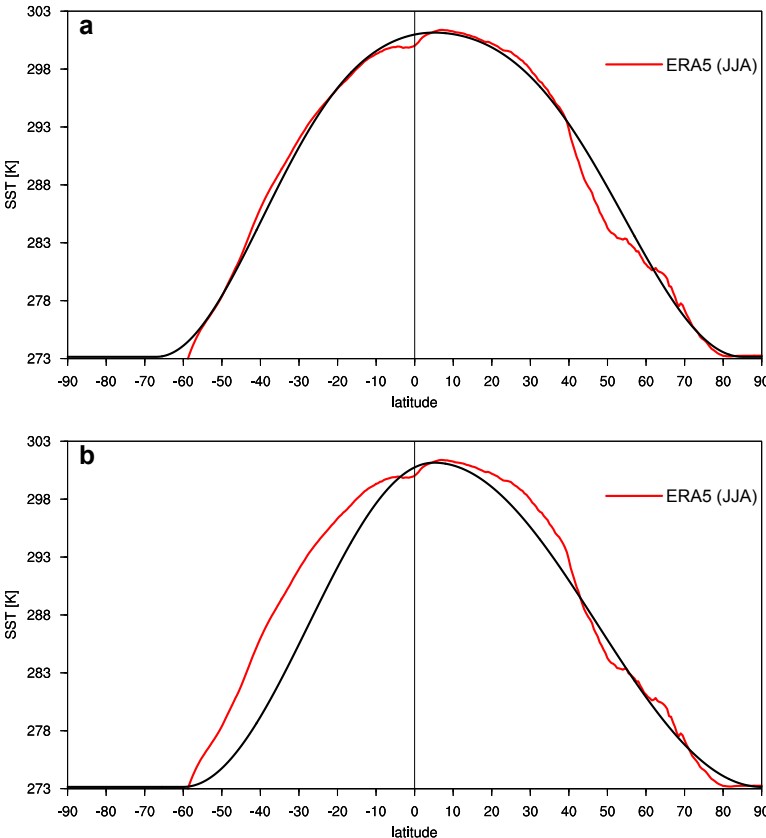

**Figure 1.** Zonal mean SST distribution based on (red) ERA5 June-August data and (black) the generalised "Qobs" distribution defined in Eq. 3 using (a) $w_1 = 0.5$, $\phi_{max} = 5°$, and $\phi_0 = 67°$ in the SH and $\phi_0 = 85°$ in the NH (this setup yields a double ITCZ). (b) For $w_1 = 1$, $\phi_{max} = 5°$, and $\phi_0 = 60°$ in the SH and $\phi_0 = 90°$ in the NH (yielding a single ITCZ).

of the changed solar orbital on the mean climate of the aquaplanet is negligible. For simulations with land surface, setting the radiation to solstice in the warmer hemisphere is crucial to obtain realistic winter and summer hemispheres but in the here used setup with fixed SSTs radiation cannot change surface temperatures.

*Mean climate*: The zonal average of selected variables for the ICON simulation and ERA5 (JJA; 1979–2022) is shown in
Fig. 2. Figure 2a corresponds to the mean climate based on the flatter SST distribution depicted in Fig. 1a. The midlatitude jet is as in observations weaker in the summer hemisphere and located farther poleward than the stronger jet in the wintertime hemisphere. The vertical motion indicates the development of a double ITCZ, with the stronger branch near the equator and the weaker branch at 10°N. Thus, the maximum ascent flanks the maximum SSTs located at 5°. The descending subtropical branch is stronger in the colder hemisphere, and a well-developed region of ascent at 45°S indicates the existence of a storm
track that is more intense in the SH compared to that in the summer NH, which is also seen in vertically averaged eddy kinetic energy (Fig.3). A double ITCZ is a feature observed in reanalysis data, for example, over the western Pacific (Fig. 2c). The





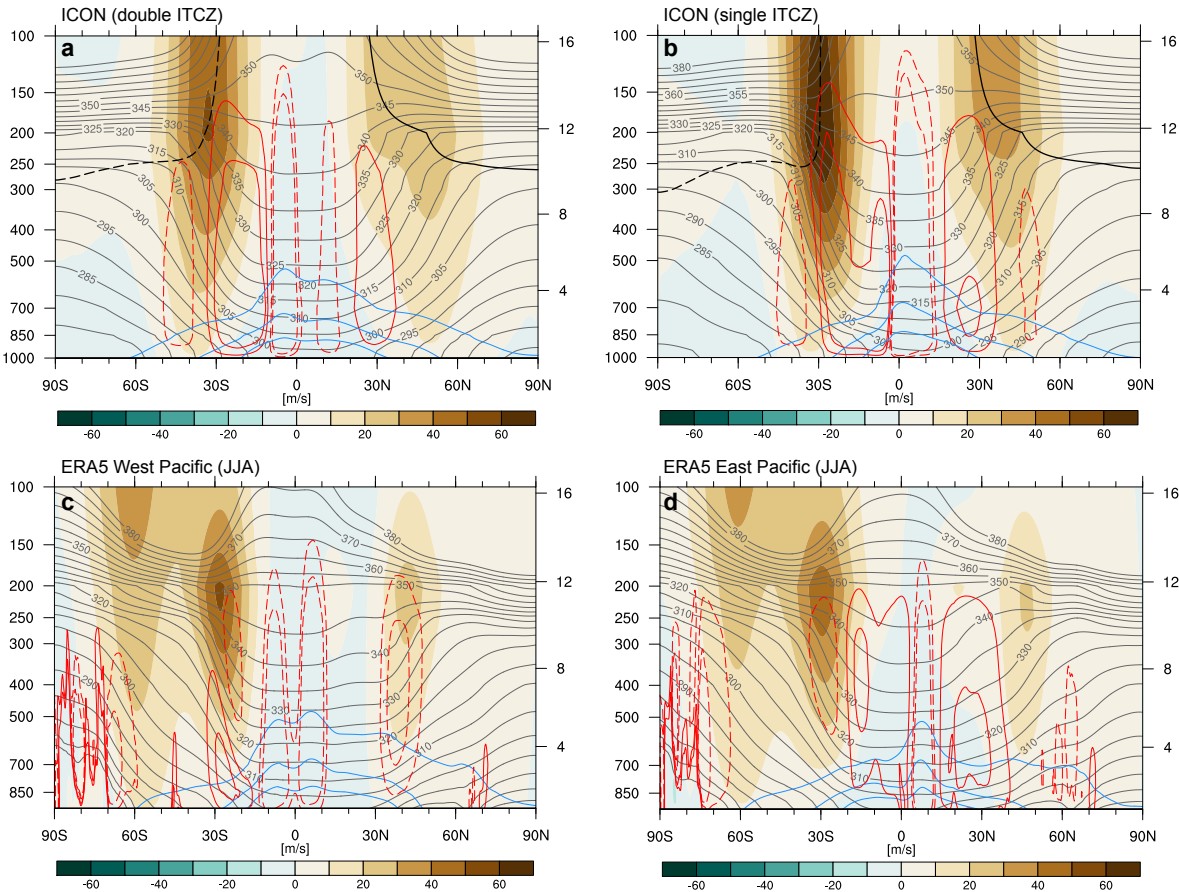

**Figure 2.** Zonal mean zonal wind (shading; m s$^{-1}$), potential temperature (gray contour; in steps of 5 K), vertical motion (red contours; upward dashed and downward solid; $\pm$ 0.4 and 0.8 hPa h$^{-1}$), specific humidity (blue shading; 3, 7 and 11 g kg$^{-1}$) for ICON aquaplanet simulations with (a) the flatt and (b) the steep SST distributions shown in Fig. 1. (c,d) Similar as (a,b) but for ERA5 averaged across the (c) West Pacific (160°E–180°) and the (d) East Pacific (120°W–180°) during JJA. Additionally shown in (a,b) is the dynamical tropopause (black dashed and solid; $\pm$2 PVU).

second, steeper SST distribution (Fig. 1b) results in the formation of a single ITCZ (Fig. 2b), as it is observed, for example, over the eastern Pacific (Fig. 2d). In the single ITCZ case a single precipitation peak emerges north of the equator in the warmer hemisphere, while in the double ITCZ case the precipitation peak in the warmer hemisphere is reduced and a second and stronger precipitation peak emerges slightly south of the equator (Fig.3b, d), which is in agreement with patterns of vertical motion (Fig. 2a). Precipitation peaks are flanked by two descending subtropical branches, both of which are characterised by almost zero precipitation in the winter hemisphere but not in the warmer hemisphere (see zonal mean next to Fig.3b, d). The



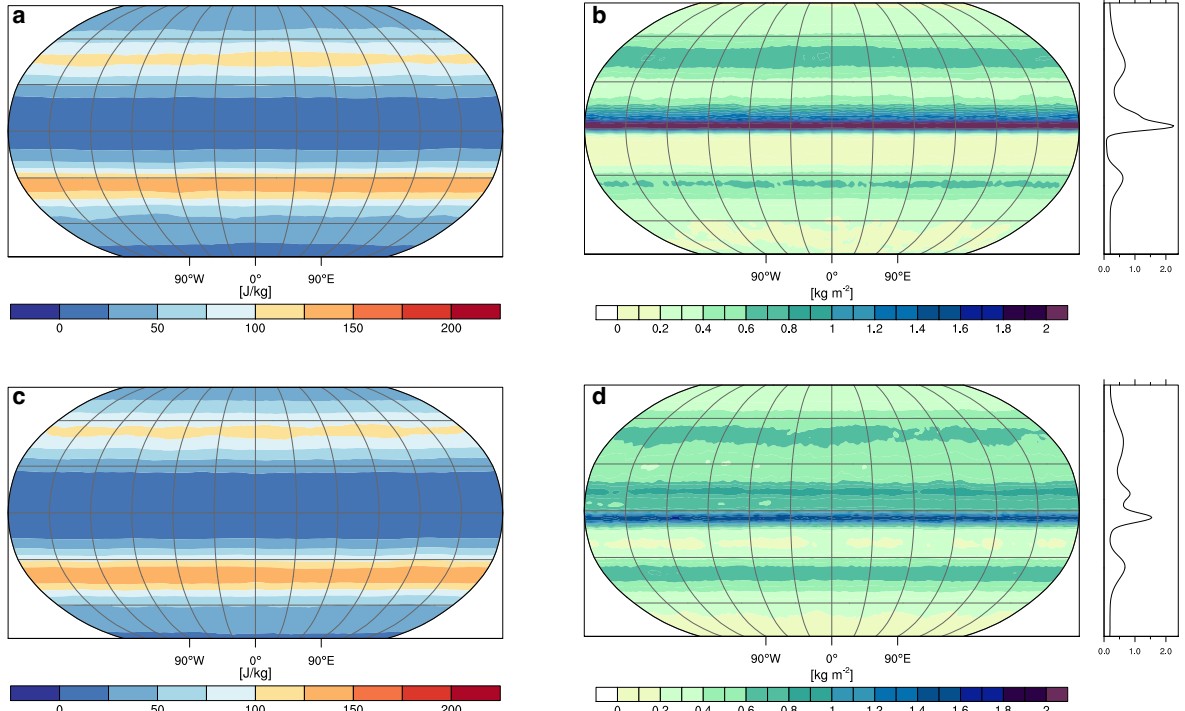

**Figure 3.** (a,c) 10-day highpass filtered eddy kinetic energy (EKE; J kg$^{-1}$) vertically averaged between 100–850 hPa for (a) the single and (c) the double ITCZ base state. (b,d) 6-hourly accumulated precipitation (shading; kg m$^{-2}$) for (b) the single and (d) the double ITCZ base state.

jet streams are stronger in both hemispheres compared to the flat SST case, as expected due to the locally larger meridional SST gradients, and the winter jet reaches up to 60 m s$^{-1}$ (Fig. 2a, b), in agreement with reanalysis data (Fig. 2c, d).

Overall, the asymmetric SSTs result in a summer and winter hemisphere with realistic looking large-scale flow features such as midlatitude jets, storm tracks and precipitation patterns. Because double and a single ITCZ are both observed in reanalysis data (albeit at different longitudes), both SST configurations are used hereafter to examine the response of the circulation in the summer hemisphere to a uniform 4 K warming.



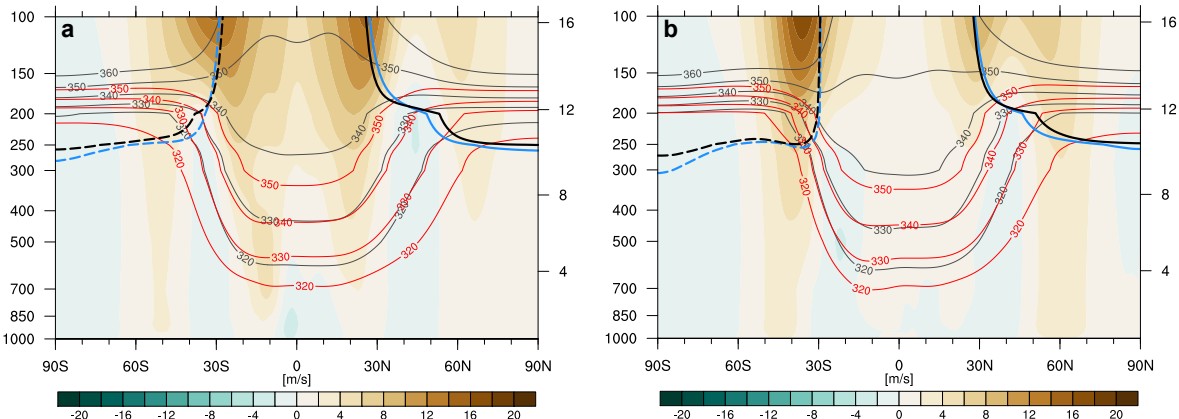

**Figure 4.** Zonal mean difference between control and simulation warmed at the surface by +4 K for the (a) double and (b) single ITCZ simulation. Shown are differences in zonal wind (shading; m s$^{-1}$), selected isentropic surfaces (red warmed and gray control simulation) and the dynamical tropopause (blue control and black warmed simulation).

## 4 Circulation response to uniform warming

### 4.1 Jet stream waviness

First, we note that the climatological waviness varies vertically and increases towards the surface (Fig. 5a), reflecting the fact that the jet becomes more zonal at upper levels. For example, in the single ITCZ run, the median jet waviness decreases in the summer hemisphere by approximately 12 % when moving from 330 to 340 Kelvin. The reduction with height is in a similar order of magnitude for the winter hemisphere, though the waviness is in general smaller in the winter hemisphere by approximately 40 % on 330 and 340 K (Fig. 5a).

Next we examine waviness changes between warmed and control simulations. When comparing the jet waviness on similar isentropic surfaces in the summer hemisphere, we find an increase in jet waviness for example on 330 K (Fig. 5b). In this case, the distribution of the jet waviness in the warmed simulation is shifted toward higher values by about half a standard deviation of the control run and consequently reaches higher extreme values (yellow in Fig. 5b). The conclusion from such a comparison would be that the jet waviness increases in a warmer climate, but the change is in the same order as the change we observe when comparing waviness on the 330 K and 340 K surfaces in the control run. Hence, if jet waviness is compared on a constant surface, changes in the height of that surface introduce a spurious trend that likely occure from a change in the height of that surface. This result is in agreement with the findings of Barnes (2013).

A contrasting picture emerges when jet waviness is compared between two isentropic surface that are on the same height in pressure space in the zonal mean of the control and warmed simulation. Fig. 4 shows the height of the 320, 330 and 340 K isentropic surfaces in the control and warmed simulations. For example, the height of the 330-K surface in the control simulation corresponds well to the 340-K surface in the warmed simulation and the same is true for the 320 and 330-K surfaces.





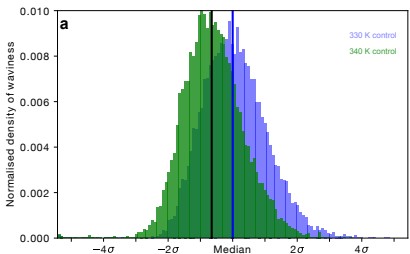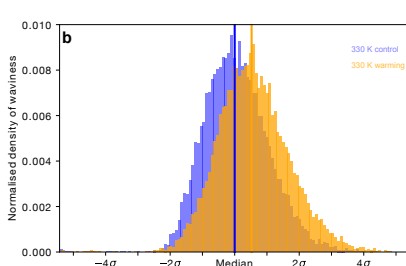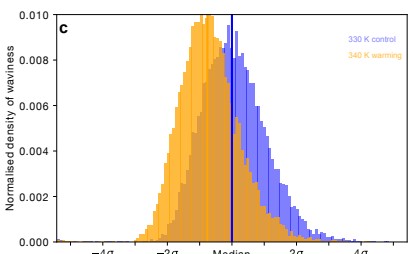

**Figure 5.** Normalised histograms of 6-hourly jet stream waviness for the summer hemisphere on (a) 330 (blue) and 340 K (green) in the control simulation; (b) 330 K in the control (blue) and warmed (orange) simulations; (c) 330 K in the control (blue) and 340 K of the warmed (orange) simulations in the single ITCZ run. Vertical lines indicate the median of the respective distributions.

While the slope of the isentropes does not change at mid-latitudes, the changes in altitude in response to a 4 K surface warming lead to a match in height between isentropic surfaces that are 10 K apart. Note that for very high surfaces, such as 350 K, the
upward expansion of diabatic heating in the tropics causes the isosurface to bend downward in the warmed simulation, while it bends upward in the control simulation. Comparing jet waviness on the 330 K and 340 K surfaces, the distribution of the warmed simulation is shifted toward lower values compared to the control simulation (Fig. 5c), and the conclusion is that the jet waviness decreases in the warmer simulation. That is, by adjusting the isentropic level, the trend reverses.

Finally, as in Martin (2021), we compute the jet waviness at each 6-hourly time step on the isentrope with maximum wind
speed and also find a systematic shift toward reduced waviness (Fig. A2). The median values computed based on the 6-hourly time series of isentropes with maximum wind speed is 333.4 K for the control simulation with single ITCZ and 344.2 K for the corresponding warmed simulation. The above comparison of the jet waviness between the 330 K and 340 K is thus reasonable as both are close to the mean isentrope that carries maximum wind speed. The increase in jet waviness found by Martin (2021) is thus not reproduced in zonally uniform APE with uniform warming, at least not on the summer hemispheric-side of the APE.

On the colder hemisphere, the picture is more complex. Accounting for the change in the altitude, jet waviness on 320 K is according to the climatological zonal mean best compared to 330 K in the warmer simulation. The jet waviness also reduces (Fig. A1a). Further above, however, when comparing 330 K to 340 K, we find almost similar distributions of the jet waviness (not shown). Even further above, for the 340 K control compared to the 350 K warmed case, we find an increase in jet waviness (Fig. A1b). Accordingly, jet waviness at this level appears to increase and it does not increase due to a change in the altitude
of the isentropes. We come back to this result after discussing changes in the amplitude of small and large waves in the next section.

As a side note, in the above section, consideration was given to the changes in jet waviness in the single ITCZ simulation. Qualitatively the results are similar for the the double ITCZ run. The results are hence not sensitive to this aspect of the mean state.




**Figure 6.** Shading depicts amplitude differences $\Delta A_k(p,\phi)$ for $k = 3, 5, 7,$ and 9 waves in (a,c,e,g) the single ITCZ and (b,d,f,h) the double ITCZ runs, respecively. Light black contours show the (time averaged) amplitudes $A_k(p,\phi)$ of the control simulation for the respective wavenumbers with a contour spacing of $1 \, \mathrm{m \, s^{-1}}$. Purple dashed and black solid contours show the zonal wind (starting at $10 \, \mathrm{m \, s^{-1}}$, in $10 \, \mathrm{m \, s^{-1}}$ steps) and the zonal mean $2 \, \mathrm{PVU}$ line, respectively.

## 4.2 Changes in Rossby wave amplitudes

In this subsection we further examine Rossby wave characteristics in our simulations, by considering changing amplitudes of different wavenumber waves. Hereby, we examine specifically, whether or not the "large-get-stronger, small-get-weaker" wave response to warming (Chemke and Ming, 2020) is apparent in our simulations. Figure 6 shows the climatological amplitudes $A_k(p,\phi)$ as well as amplitude differences $\Delta A_k(p,\phi)$ for wavenumbers $k = 3, 5, 7$ and 9 for both the single (a,c,e,g) and double (b,d,f,h) ITCZ setups (results for wavenumbers $k = 4, 6, 8$ and 10 are qualitatively similar, not shown). Evidently, the response



to uniform warming differs considerably between large waves ($k = 3, 5$) and synoptic wavenumber waves ($k = 7, 9$), but is qualitatively very similar between the single and double ITCZ setups. In the summer hemisphere, wavenumber 3 and 5 waves feature largest amplitudes pole-ward of the jet and amplify under warming. However, this amplification is largely constrained to levels above 300 hPa and, moreover, is largest at the pole-ward and equator-ward edges of the jet. This increase in wave

amplitude does not project on an increase in jet waviness at the jet axis (as shown in the previous section), likely because it occurs away from the jet axis and is offset by the decrease in wave amplitude of short waves. The increase in amplitude of large waves thus essentially constitutes an upward extension of the regions with largest wave 3 and 5 amplitudes. In stark contrast, the region of maximum amplitude of waves 7 and 9 is located close to the jet maximum in the summer hemisphere and the $\Delta A_7(p, \phi)$ and $\Delta A_9(p, \phi)$ fields reveal a decrease in wave amplitude below the jet maximum as well as on its equator-ward

side. At the same time, a weak increase in these wavenumber waves' amplitudes is discernible above 200 hPa, in particular on the pole-ward side of the jet.

The changes in wave amplitudes in our APE indeed reveal a "large-get-stronger, small-get-weaker" wave response to warming, and are thus consistent with results of previous studies assessing such changes in far more complex global warming simulations (Rivière, 2011; Chemke and Ming, 2020). One line of reasoning for explaining this response is that large waves

are much less confined to lower levels than small waves and therefore benefit from an increase in upper-level baroclinicity, whereas an increase in low-level baroclinicity would benefit the growth of both small and large waves (Rivière, 2011). A closer look at the changes in the Eady growth rate shows indeed an increase in the Eady growth rate at 20–30°N/S between 250–100 hPa (not shown, but note the steeper slope in this region in Fig. 4), consistent with the arguments presented in Rivière (2011). However, given that in many parts of the free troposphere baroclinicity changes little in our simulations it is currently not

entirely clear what causes the reduction in the amplitude of synoptic wavenumber waves in the free troposphere.

As mentioned before, the increase in wave amplitude of large waves seems not to be reflected in an increase in jet waviness. But there is one exception. The increase in the amplitude in the colder hemisphere for wave numbers 5 (blue shading in Fig. 6c) is strongest close to where an increase in the waviness of the jet was observed at very high isentropes (350 K). While in general changes in all wave numbers collectively drive the waviness of the jet contour, in this sector the response seems

to be dominated by changes in wave number 5 (and less), while elsewhere it is dominated by the decrease in amplitude (and hence jet waviness) of wave number 7 (and larger). The changes in the wave numbers that dominate the signal at different levels are all in agreement with the previously reported waviness changes. While those in the core are dominated by synoptic wave numbers, there is apparently also an increase in planetary wave amplitude and jet waviness that occurs in the absence of zonal asymmetries (and AA), which is a key requirement for the mechanism described in the model of Moon et al. (2022).

Also from the above it follows that synoptic baroclinic waves and changes in their amplitude dominate at the jet stream level and thus cannot be neglected.

In summary, Fig. 6 clearly reveals that the "large-get-stronger, small-get-weaker" wave response to warming is apparent in our APE simulations. This suggests that this response, first diagnosed in fully coupled GCMs, may be unrelated to changes in zonal asymmetries, such as land-ocean contrast, but instead arises due changes that are reproduced in our simulations, such

as changes in free tropospheric and upper-level baroclinicity. Furthermore, Fig. 6 further underlines that changes in wave





amplitudes are not only scale-dependent (Chemke and Ming, 2020) but also varies between altitudes as well as latitudinally (Wills et al., 2019), even in absence of zonal asymmetries. Moreover, on different levels the changes in wave amplitude can be reconciled with changes in jet waviness. However, in the jet core, synoptic wave numbers dominate the changes.

### 4.3 Preferred phasing of high-amplitude waves

Recently Kornhuber et al. (2020) found that in Northern Hemisphere summer quasi-stationary, synoptic wavenumber waves exhibit a preferred phasing that fosters concurrent heat waves in major breadbasket regions, in particular central North America, western Europe and western Asia. Such slow-moving amplified hemispheric wave patterns have been observed during the concurring weather extremes in early summer 2018 (Kornhuber et al., 2019) and their increasingly frequent occurrence has been suggested as potential cause of the significantly larger trends in heat waves in western Europe compared to other regions

(Rousi et al., 2022). For planetary-scale stationary waves, i.e., $k = 1, ..., 4$, it is well understood that they have a preferred phasing set by quasi-stationary forcing mechanisms, such as land-ocean contrasts, topography or slowly evolving sea surface temperature anomalies (e.g., Hoskins and Woollings, 2015). However, for synoptic wavenumber ($k > 4$) waves the causes of a preferred phasing of high-amplitude waves are far less clear. Previously formulated hypothesis based on the notion of quasi-resonant amplification (e.g., Petoukhov et al., 2016) have significant shortcomings on a theoretical level (Wirth, 2020; Wirth

and Polster, 2021), and other explanations for this phenomenon such as the notion of recurrent Rossby waves (Röthlisberger et al., 2019) remain rather observational. To narrow-down the possible causes of a preferred phasing of high-amplitude synoptic waves, we thus assess whether or not a preferred phasing is apparent in any of our APE simulations.

Figure 7 shows histograms of the observed phase during weeks with high-amplitude waves for wavenumbers 5–8 (columns) and all four simulations (rows). As in Kornhuber et al. (2020), for some wavenumbers and simulations there are rather large

differences in the phase histograms for high-amplitude waves compared to climatology (compare red and black lines in Fig. 7), most notably so for wavenumber 8 in the double ITCZ warming and single ITCZ control simulations. Nevertheless, the deviations between the histograms for high-amplitude weeks in almost all cases lie within the 2.5th to 97.5th percentile range of the bootstrapping samples. This suggest that similarly large deviations can arise due to sampling uncertainty, i.e., because the sample of weeks with high-amplitude waves (36–47 values in this study, 44–47 values in Kornhuber et al. (2020)) is rather

small. The few exceptions mentioned above (e.g. Fig 7h) should also not be over interpreted, as we perform multiple statistical hypothesis tests here (each at a significance level of 5%), and thus some erroneous rejections of our null hypothesis (no preferred phasing) are expected.

That is, contrary to the "large-get-stronger, small-get-weaker" wave response discussed above, where a Rossby wave response to warming previously diagnosed in far more complex GCM simulations is indeed apparent in our APE simulations,

we find no evidence for a preferred phasing of high-amplitude, quasi-stationary synoptic wavenumber waves in our simulations. This suggests that, similarly to quasi-stationary waves, also for synoptic wavenumber waves a preferred phasing of high-amplitude waves in the real atmosphere is likely related to quasi-stationary forcing mechanisms and zonal asymmetries, such as topography and land-sea contrasts.



**Figure 7.** Weighted histogram of phase positions for wavenumber 5 (a,e,i,m), 6 (b, f, j, n), 7 (c, g, k, o) and 8 (d, h, l, p) for the double ITCZ control simulation (first row), the double ITCZ warming simulation (second row), the single ITCZ control simulation (third row) and the single ITCZ warming simulation (fourth row). The red line shows the histogram for the $N_k$ high-amplitude weeks, the gray shading and the black line show the 2.5th to 97.5th percentile range and the median for the randomly selected sets of weeks, respectively. The histograms have been constructed using eight bins with $\pi/4$ width.





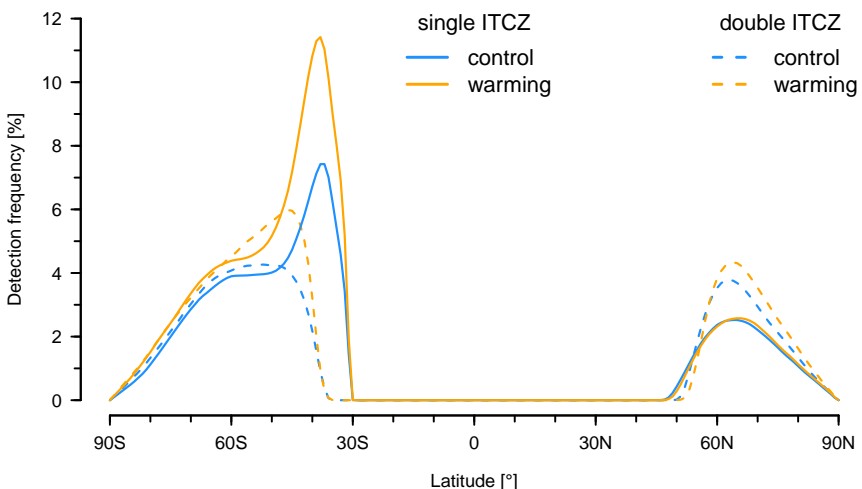

**Figure 8.** Zonal mean detection frequency of blocks (see methods) weighted by $\cos(\mathrm{latitude})$ for control (blue; solid) and warmed (orange; solid) single ITCZ and control (blue; dashed) and warmed (orange; dashed) double ITCZ simulations. Blocking detection is based on $\mathrm{VAPV_0}$ and $\mathrm{VAPV_A}$ for the control and warmed simulations, respectively.

## 4.4 Changes in blocking

In the control simulation, zonal mean frequencies of blocking based on VAPV0 of around 4 % are found in the winter hemisphere's midlatitudes between 45°S and 70°S (Fig. 8). The single ITCZ simulation further features a pronounced and highly localized maximum in the subtropics reaching frequencies of around 8 %. This maximum, which is not present in the simulation with a double ITCZ, coincides with the intense subtropical jetstream (Fig. 2b) and the associated steep climatological PV gradient (not shown). In the summer hemisphere, notable blocking frequencies are confined to poleward of 50°N with a

maximum in both simulations located north of 60°N. While in the double ITCZ simulation the maximum frequency is comparable between winter and summer, the hemispheric average is considerably lower in summer due to the poleward shift of blocking occurrence. In the single ITCZ simulation, the maximum frequency reaches only about half of that in the double ITCZ simulation.

   As pointed out by Croci-Maspoli et al. (2007), changes in the tropopause height can potentially cause spurious trends

in blocking frequency. In the warmed simulations, the tropopause is shifted upward by 10-20 hPa (Fig. 4), which results in a reduction of the climatological $\mathrm{VAPV_0}$ as compared to the control simulations (cf. circles in Fig. 9). Using $\mathrm{VAPV_0}$ for the identification of blocks in the warmed simulations results in a reduction of hemispheric blocking frequencies in summer, as well as in the case of the double ITCZ simulation also in winter. This change in blocking occurrence under warming is consistent with the spurious change expected due to a rise of the tropopause and an associated reduction of climatological $\mathrm{VAPV_0}$. To

test for this possibility, we additionally performed the blocking identification in the warmed simulations with layers shifted





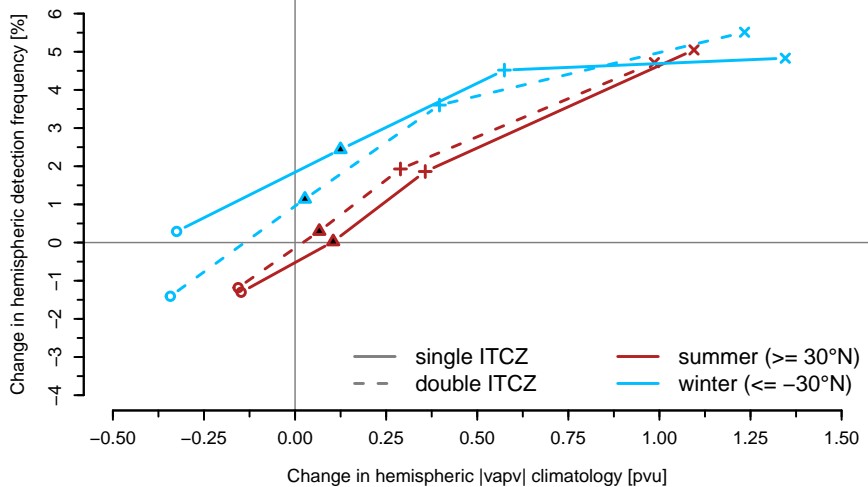

**Figure 9.** Sensitivity of changes in blocking frequency to vertical shifts of detection layer. The plot shows the change in hemispheric (poleward of 30°N or 30°S) VAPV climatology (x-axis) and detection frequency of blocks (y-axis) in warmed vs. control simulations with single ITCZ (solid) and double ITCZ (dashed). Red and blue indicate averages taken over the summer ($> 30°N$) and winter hemisphere ($> 30°S$), respectively. Blocking detection in control simulations is based on $VAPV_0$, whereas in warmed simulations it is based on four different VAPV definitions: $VAPV_0$ (circle), $VAPV_{25}$ (plus), $VAPV_{50}$ (cross), and $VAPV_A$ (black filled triangle).

upward by 25 hPa and 50 hPa, i.e., $VAPV_{25}$ and $VAPV_{50}$, respectively, thereby compensating for the effect of the elevated tropopause. Using $VAPV_{25}$ or $VAPV_{50}$ results in a positive change of climatological VAPV as compared to the control, along with an increase of the hemispheric blocking frequency (plus and crosses Fig. 9) instead of a decrease. This points towards a high sensitivity of blocking detection to the choice of vertical layer and underlines that care must be exercised when comparing

changes of blocking occurrence in such simulations of a warming climate.

One could argue that VAPV used for blocking identification should not change under warming. This is approximately true for $VAPV_A$ (black filled triangles in Fig. 9). In this case, one finds that under warming zonal mean blocking frequencies remain nearly unchanged in the summer hemisphere, whereas they increase in the winter hemisphere (Fig. 8). The latter increase is largest in the subtropics in the single ITCZ simulation and goes along with the intensification of the subtropical jetstream

(Fig. 4b).

## 4.5 Changes in surface cyclones

Feature-based detection of cyclones provides an opportunity to study the cumulative impact as well as the individual life cycle characteristics and changes of surface cyclones. In both simulations, the frequencies of detection of cyclones are greater in the winter hemisphere (Fig. 10). Cyclone detection frequencies indicate the fraction of time steps a region is affected by a





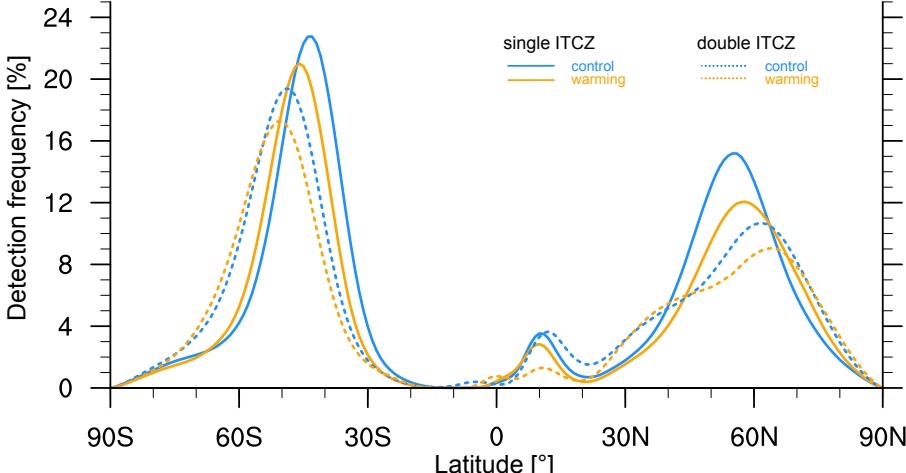

**Figure 10.** Zonal mean cyclone frequencies weighted by $\cos(\mathrm{latitude})$ for (a) control (blue; solid) and warmed (orange; solid) single ITCZ and (b) control (blue; dashed) and warmed (orange; dashed) double ITCZ simulations.

cyclone and range from zero (no cyclones) to 100 % (a cyclone is present at every time step). The largest frequencies in the zonal mean are found near 40–45°S in the single ITCZ simulation (solid blue in Fig. 10). The storm tracks are in the single ITCZ simulation as expected located closer to the equator in both hemispheres if compared to the double ITCZ simulation. Consistent with observation, the summertime storm tracks are located more poleward compared to the wintertime storm tracks.

In the warmer simulations, peak cyclone frequencies reduce in both hemispheres and both simulations types. Several changes
in life cycle characteristics can contribute to a change in the cyclone frequencies, for example, a change in the number of cyclones (i.e., this tends to reduce the local detection frequencies), their propagation speed (i.e., high propagation speed causes locally a reduced cyclone frequency) in combination with a reduction of their life time, which tends to reduce the local detection frequencies. Table 1 summarises the number of identified surface cyclone tracks and their changes under warming. In agreement with previous studies (Sinclair et al., 2020; Schemm et al., 2022), the number of unique cyclones decreases in both simulation
under warming. The reduction is of comparable order in the winter and summer hemispheres, for example, in the double ITCZ run the reduction in cyclone numbers is in the order of 10–11 %. It is also of similar order in the single ITCZ run and in both hemispheres. Hence, cyclone numbers reduced in general by about 10 % in this APE. Indeed, Zappa et al. (2013) and Chang et al. (2016) identify a reduction in the number of summertime extratropical cyclones in reanalysis data in the order of 4 % per decade over the entire North Hemisphere and regionally up to 10 % for example over North America (Chang et al., 2016). The
decrease in summertime extratropical cyclone activity is projected to continue into the future (Zappa et al., 2013; Chang et al., 2016).

Changes in the lifetime of surface cyclone tracks are considered. In agreement with previous studies (Schemm et al., 2022), the lifetime tends to decrease in both hemispheres. The changes are calculated based on a histogram of binned lifetimes, starting with lifetimes between 1 and 2 days and continuing in increments of 24 hours to lifetimes of more than two weeks.





**Table 1.** Number of surface cyclone tracks and change relative to the control run in both simulation types and hemispheres (NH corresponds to the summer and SH to the winter hemisphere).

|  | Total (CTRL) | Total (WARM) | NH (CTRL) | NH (WARM) | SH (CTRL) | SH (WARM) |
|---|---|---|---|---|---|---|
| Single ITCZ | 35'899 | 32'379 (≈-10%) | 18'723 | 17'103 (≈-9%) | 17'174 | 15'276 (≈-11%) |
| Double ITCZ | 35'717 | 31'975 (≈-10%) | 18'485 | 16'487 (≈-11%) | 17'232 | 15'488 (≈-10%) |

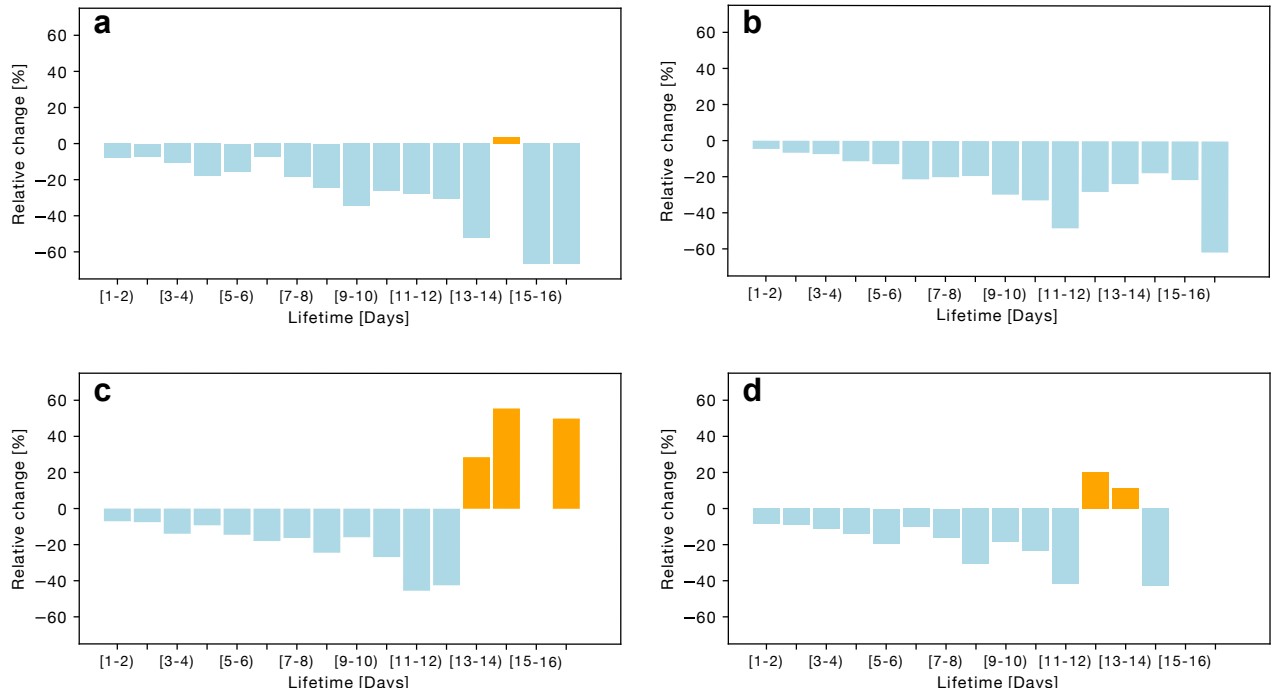

**Figure 11.** Change in lifetime of surface cyclone tracks binned according to different lifetimes in days for (a,c) the double and (b,d) single ITCZ runs and for the summer (a,b) and (c,d) winter hemispheres.

The relative changes are then derived from the number of binned cases in the control and warmed simulations. The reduction appears to be greatest for the long-lived cyclones in the summer hemisphere (Fig. 11a,b), but the large numbers also result from the small number of cases in the very long-lived cyclones. In the winter hemisphere (Fig. 11c,d) most lifetimes exhibit a similar decrease, except for the very long lived cyclones, which indicate even an increase. The results confirm the findings in Schemm et al. (2022) who identified an increase in lifetime in particular for cyclones downstream of an additional SST front, likely due

to an extension of the life cycle by diabatic processes. A drawback is the low numbers of long-lived cyclones and presumably a sensitivity of the result to the cyclone identification and tracking. Hence, we consider the increase for the long-lived cyclone to be less robust compared to the decrease for the short-lived cyclones (up to 7'000 cases in a hemisphere have a lifetime of 1–2 days compared to approximately 100 cases with a life time of more than 10 days).





## 5    Conclusions

This study introduces a simplified aquaplanet setup with winter and summer hemispheres to investigate the response of the circulation to uniform warming in both hemispheres. Following the long tradition of previous aquaplanet experiments, a variant of the well-established zonally uniform "Qobs" SST distribution is used that allows to mimic zonal mean June-July-August (JJA) SSTs as observed in reanalysis data. This SST distribution allows also to control the formation of a double and single ITCZ as observed over the East and Central Pacific, respectively, depending on the cross-equatorial SST gradient. A 10-year

simulation with radiation fixed at equinox results in realistic looking large-scale circulation features such as jet streams and storm tracks, which are less intense and more poleward in the summer hemisphere compared to their counterparts in the winter hemisphere, which is in agreement with observations. The setup serves to explore some challenging research questions related to changes in circulation with global warming, which we summarise below.

*Jet waviness*: The waviness of the jet stream in a warmer climate has received much attention because of its association with

extreme weather events (temperature and hydrological extremes) (e.g., Francis and Vavrus, 2015; Blackport and Screen, 2020). We choose a geometric definition of jet waviness based on the 2 PVU contour (Röthlisberger et al., 2016). We first note that the waviness in the atmosphere increases downwards. Therefore, if we examine the jet waviness at similarly valued isentropes (e.g. 320 K) in the control and warmed simulations, we find that the jet waviness increases. This spurious trend results from the fact that in the warmed simulation the corresponding isentropes are closer to the surface. This is in agreement with arguments

presented by Barnes (2013). Next, we compared the jet waviness on isentropes that are at the same height in pressure space identified from the zonal mean climatology. With this approach, we find that the jet waviness decreases. Finally, we compared the distribution of jet waviness when calculated at each time step on the isentrope with the highest wind speed (Martin, 2021). Again, the jet waviness decreases. There is only one exception, and that is at very high isentropes (e.g. 350 K) in the colder hemisphere, where the jet waviness increases and is not due to a change in the height of the isentrope (see also below discussion

of change in wave amplitude for different wave numbers). Overall, the observed upward trend found by Martin (2021) is not reproduced in our zonally symmetric simulations of uniform warming, but based on our findings a sensible way forward is – as in Martin (2021) – to compare jet waviness on isentropes located at the jet maximum and not on isentropes or pressure surfaces hold constant over time.

*Rossby wave amplitude*: A key result of our study is that our simulations, which feature no zonal asymmetries such as

land-sea contrasts, topography etc., reproduce the "large-get-stronger, small-get-weaker" wave response to warming that was found by Chemke and Ming (2020) in fully coupled GCMs. The decrease in amplitude of small waves (wave numbers 7 and above) is greatest at jet stream levels and slightly equatorward of the mean jet position. This is consistent with the tendency of short waves to break cyclonically (Rivière, 2011). The increase in amplitude of large waves (wave numbers 5 and smaller) is found mostly, but not exclusively, at upper stratospheric levels and within some distance of the jet stream. The increase

tends to be stronger on the poleward side of the mean jet position, which is in agreement with the tendency of long waves to break anticyclonically (Rivière, 2011). The changes are found in both hemispheres, but they are more pronounced in the colder hemisphere. In the colder hemisphere, the increase in the amplitude of long waves at higher altitudes (i.e., 350 K) occurs




near the location where we find an increase in the waveiness of the jet. The reduction in jet waviness at lower isentropes (i.e.,
320–330 K) and at the height of maximum wind speed is consistent with the reduction in synoptic wave amplitude, which
dominates the change in jet height. Note, the net waviness results from a combination of changes in all wavenumbers, but
short waves contribute by a wider margin to the geometric jet waviness (i.e., due to more troughs and ridges and thus more
latitudinal variations of the 2 PVU contour for short waves compared to long waves along a fixed latitute circle). The increase
in long wave amplitude does not require zonal asymmetries as in the model of Moon et al. (2022), which also neglects synoptic
baroclinic wave growth, which in our case dominate the wave amplitude changes at the height of the jet stream.

*Rossby wave phase*: During so-called high-amplitude wave events, Kornhuber et al. (2020) identify a preferred phasing
of wavenumbers 5 and 7 in the Northern Hemisphere. While high-amplitude wave events are sometimes associated with
concurring weather extreme events (Kornhuber et al., 2019; Rousi et al., 2022), the causes of this preferred phasing is currently
unclear. Here, lacking topography and land-sea contrast, we cannot identify a preferred phase of waves numbered 5–8 during
weeks with anomalously large wave amplitudes, neither in the colder nor in the warmer hemisphere. This is not too surprising,
because our simulations do not contain any zonally asymmetric wave forcing mechanisms, which could anchor the phase of
synoptic (or longer) waves (White et al., 2021). We nevertheless explicitly present these results here, in order to motivate
future studies on the causes of a preferred phasing of high-amplitude synoptic wavenumber waves based on our APE setup. In
particular, we envisage that our APE setup could be expanded, e.g., by adding idealized topography, SST fronts or comparable,
in order to gain a mechanistic understanding of how this phenomenon comes about, and to investigate to which extent forms
of wave resonance might be involved (Charney and Eliassen, 1949; Petoukhov et al., 2016; Wirth, 2020; Wirth and Polster,
2021).

*Blocks*: Among all diagnostics used in this study, blocking detection based on PV vertically averaged between 500–100 hPa
is the most sensitive to uniform warming and the associated rise of the tropopause. A spurious reduction in blocking frequencies
results from the heightened tropopause in the warmer simulation (as already noted by Croci-Maspoli et al., 2007). A simple
upward adjustment of the VAPV in steps of 25 hPa, which accounts roughly for the tropopause lift, leads to an increase in
detection frequencies, but is also associated with an over-proportionally large increase in VAPV, potentially explaining the
larger blocking frequencies. Once VAPV is very carefully tuned so that hemispheric-wide VAPV remains unchanged between
the control and warmed simulations, blocking frequencies remain also unchanged in summer but increase in winter. These
results suggest that, rather than actual dynamical changes, the identified blocking frequency changes are predominantly a
function of changes in the hemispheric VAPV in the vertical layer that is considered in the detection algorithm. Our confidence
in the above summarised blocking frequency changes thus remains low, even after the required adjustment. We therefore
advise increased caution with trends derived from blocking detection schemes that are either directly or indirectly a function of
pressure. Note that according to the review by Woollings et al. (2018), there is evidence of a reduction in blocking with global
warming from a variety of detection algorithms and models. However, as some of these algorithms are pressure dependent it is
unclear to which extent this reduction is affected by vertical shifts in the tropopause, and should be systematically investigated.

*Surface cyclones*: As noted already in previous aquaplanet studies (Sinclair et al., 2020; Schemm et al., 2022), the number
of cyclones decreases under warming. Here it reduces globally by about 10 % and in the colder and warmer hemispheres. This



reduction is already seen in reanalyis data, for example, Chang et al. (2016) finds a reduction of up to 10 % over North America and of up to 4 % in the NH during summer since 1979 (until 2014). Zappa et al. (2013) report a projected reduction of 4 %

(2 %) during NH winter (summer) over the North Atlantic under the RCP4.5 emission scenario and this reduction is therefore a robust signal. Except for long-lived cyclones, cyclone life times decrease as well in particular in the colder hemisphere.

**Final remarks**

Our study shows that feature-based detection methods need to be carefully adapted to new climates, in particular those that depend on a vertical coordinate. Moreover, it is shown that some circulation changes reported in the literature are found already

in our zonally symmetric simulations under uniform warming, such as the change in wave amplitude of short and long waves, and the strong dependence of jet waviness trends on the exact level and vertical coordinate. Other vividly discussed phenomena, however, are absent, such as the preferred phase of certain wavenumber waves during high-amplitude wave events.

In this study we have argued and demonstrated that aquaplanet simulations such as those presented here may be useful tools for resolving questions regarding circulation changes in response to warming, predominantly for two reasons. Firstly, they allow

to study the sensitivity of weather/circulation feature diagnostics to warming in a simplified setting, which helps unravelling and understanding which trends are methodological artefacts and which indeed relate to circulation changes. Secondly, they allow constraining the palette of causes of previously reported circulation changes, and furthermore allow highlighting or excluding the importance of changes in zonal asymmetries for these circulation changes. The data from our aquaplanet simulations are made publicly available (LINK WILL BE INSERTED AFTER ACCEPTANCE) and it is hoped that they will serve as starting

point for subsequent studies on the hotly debated topic of circulation changes under global warming.

*Code availability.* The ICON model is distributed to institutions under an institutional license issued by the German Weather Service (DWD). More details can be found at $https : //code.mpimet.mpg.de/projects/iconpublic/wiki/How\_to\_obtain\_the\_model\_code$ (last access: 1 Feb 2023).

*Data availability.* The primary data used for this paper are archived at ETH Zurich's Research Collection for scientific publications and

research data under a CC-BY 4.0 license for at least the upcoming 10 years (DOI WILL BE INSERTED). ETH Zurich's Research Collection adheres to the FAIR principles.



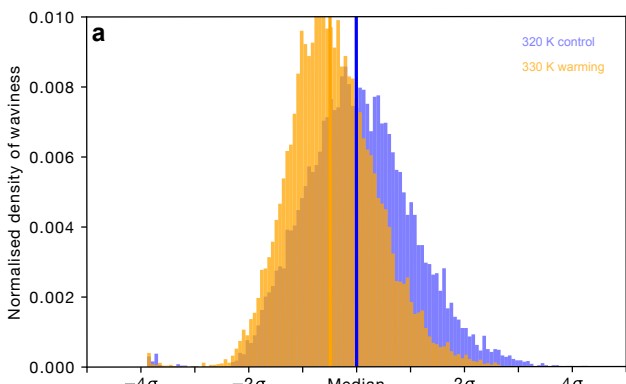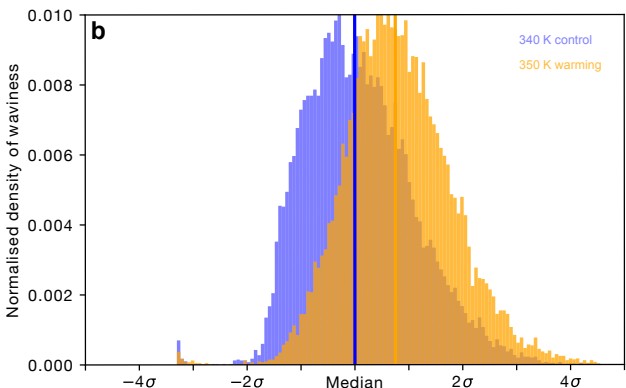

**Figure A1.** Normalised histograms of 6-hourly jet stream waviness for the winter hemisphere for (a) the 320 K and 330 K isentropes in the control and warmed simulations, respectively. (b) Similar but for the 330 K and 340 K isentropes.

## Appendix A: Additional jet waviness histograms

Figure A1 shows a normalised histograms of 6-hourly jet stream waviness for the winter hemisphere for (a) the 320 K and 330 K isentropes in the control and warmed simulations, respectively. (b) Similar but for the 330 K and 340 K isentropes.

Figure A2 shows a normalised histograms of 6-hourly jet stream waviness on the isentrope with maximum zonal wind speed for the summer hemisphere in simulation with (a) the double and (b) single ITCZ. Control is shown in blue and warmed in oranges. At the point where the histogram appears cut off, the jet waviness is zero. Median and standard deviation shown on the bottom axis are based on the waviness distribution of the control run.



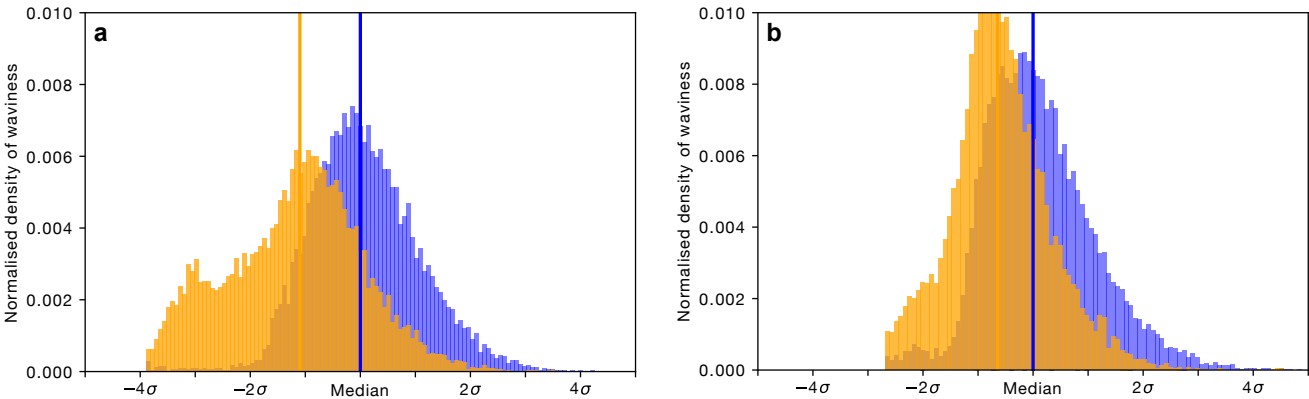

**Figure A2.** Normalised histograms of 6-hourly jet stream waviness on the isentrope with maximum zonal wind speed for the summer hemisphere in simulation with (a) the double and (b) single ITCZ. Control is shown in blue and warmed in oranges. At the point where the histogram appears cut off, the jet waviness is zero. Median and standard deviation shown on the bottom axis are based on the waviness distribution of the control run.



*Author contributions.* SeS developed the APE setup and ran the simulations. MR led the analysis of the wave amplitude and the preferred
phase position. SeS led the analyses of the waviness and the cyclone frequencies. SeS and MR led the writing of the manuscript.

*Competing interests.* Sebastian Schemm is a member of the editorial bord of Weather and Climate Dynamics.

*Acknowledgements.* The authors acknowledge the work of Lukas Papritz, who investigated the required tuning of the block detection routine
and performed the corresponding section. The Center for Climate Systems Modeling (C2SM) at ETH Zurich is acknowledged for providing
technical and scientific support. All simulations were carried out at ETH's Scientific and High Performance cluster EULER.



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
