# Peer review of "Aquaplanet simulations with winter and summer hemispheres: Model setup and circulation response to warming"

_EGUsphere, 2023_

## Author Comment (AC1)

Schemm and Röthlisberger developed a setup for aquaplanet experiments (APE), that incorporates hemispheric asymmetries (winter and summer hemisphere). They use this setup to study a range of behaviours and phenomena, mostly related to mid-latitude jet and eddy activity, in parameter regimes corresponding to "current" and "warmer" climates, as well as single and double ITCZ settings.

This is an overall interesting study nicely showing a sample of what can be done with rather simple models. Many of the aspects that are investigated relate to highly relevant and somewhat open research questions regarding climate change and general circulation behaviour. Key results include that previously reported increase in jet-waviness on a specific isentropic level in a warmer climate could be an artefact due to a vertical shift of that level and waviness at a fixed pressure might actually decrease, as well as the observation of small-scale wave weakening and large-scale waves strengthening during global warming, consistent with other studies.

The manuscript is fairly well written, the structure is clear and the figures are good. The scientific reasoning and development of conclusions seem sound. I have a few minor comments and questions for the authors, but can otherwise recommend the paper to be published.

We would like to thank the reviewer for the positive and thoughtful assessment of our manuscript. We will try to improve the reading flow when revising the manuscript.

General remarks:

- First, I have to admit that I am no expert studying APEs. The authors give information on typical APE setups (e.g. Neale and Hoskins, 2000), but I still struggle to fully understand the advantage of the hemispheric asymmetry in the "new" simulations. I feel like most of the analyses could have been performed in two separate symmetric runs (potentially with half the runtime even, keeping the computational costs equal). Is the suggested advantage simply to allow for asymmetric double ITCZ runs? In that case, I think some additional justification of why this (tropical) asymmetry should affect mid-latitude jet and eddy behaviour might be helpful. Again, maybe this only requires some further clarification.

Very good point. However, it is unlikely that this simulation would be possible in two symmetrical simulations. Symmetric winter runs are widely used. They are based on the zonal mean SST of the DJF seasonal mean in the NH, which is mirrored at the equator. Consequently, the maximum SST is at the equator and both hemispheres are symmetric. However, the summer SST used here is characterised by a poleward shift of the SST maximum. Mirroring the summer SST would result in two peaks at 5°N and 5°S and a local SST minimum at the equator, which does not seem ideal to us. The JJA SST is ideal for creating an aquaplanet with a summer and winter-like SST profile as it allows both hemispheres also to interact. A study that includes the interactions between the hemispheres (e.g., through cross-equatorial transport by the ITCZ) as the planet is warmed would not be possible with a pure summer or winter aquaplanet setup.

How the exact ITCZ structure affects the mid-latitudes is open to debate, but the subtropical jet lies along the edge of the ITCZ, its exact position is known to affect the mid-latitudes (see midwinter suppression), so one might initially expect an influence depending on the single and double ITCZ configuration. We wanted to present results for both cases, but our main finding is that the results are quite similar, which is an important take away message. From a modelling point of view, it is also very exciting that our setup allows us to control for the occurrence of both single and double ITCZs with very little change. The setup may prove useful in the future to test new model generations, for example at very high resolution, for tending to produce a single or double ITCZ.

- I think many of the figures could benefit from row and column labels. The information is mostly given in the caption, but this could help the reader take in the content.

We added row and column labels wherever we felt they were useful.

- Maybe I missed something, but I don't understand why you would expect any zonal asymmetries in terms of a preferred wave phase (Sec. 4.3)? Using purely zonally symmetric boundary conditions the mean response should (given sufficient statistics) also be perfectly symmetric, right?

  We agree with the reviewer that one does not expect a preferred phasing of high-amplitude waves in our APEs, which is confirmed by our results. The reason for nevertheless including these results are twofold: Firstly, we wanted to also include an analysis in this paper where circulation features observed in comprehensive GCMs/reanalyses are NOT reproduced by our APE simulations (contrary to the "large-get-stronger, small-get-weaker" response to warming, or the seasonal differences in the simulated storm track shifts). Secondly, even though this result could have clearly been anticipated based on basic atmospheric dynamics reasoning, the lack of a preferred phasing of high-amplitude synoptic wavenumber waves in our APE suggests that any such preferred phasing must have its physical causes in zonal asymmetries. We believe that for synoptic-wavenumber waves this point has not been made expicitly in previous literature, and our summer-like APE lends itself to underlining this in a simple and straightforward manner. For these reasons we feel it is justified to include this analysis in the paper, even though we fully agree with the reviewer that the result is not surprising. To make our initial hypothesis more explicit we now write at the beginning of Section 4.4 that we do not expect any preferred phasing of high-amplitude waves in our APE

Specific remarks:

L100: Please introduce the acronym APE

Done.

L134 "see below": Maybe better reference the specific section

We removed all "see below" statements.

Eq. 1: Since you are interested in the vertical structure, I was just wondering what happens if you normalise your Delta A_k with some density factor (or equivalently study changes in wave energy). Not sure if this leads to any insights or different conclusions, but might be worth considering.

Interesting comment, thank you! We believe your suggestion would indeed be the preferred approach if one was working with some form of wave energy or EKE (e.g., as in Chemke and Ming, 2020). Here, however, we investigate only the amplitude of different wavenumber waves in the meridional wind field. In our case weighting Delta A_k with density would not yield a quantity that is physically very easily interpretable. Also, note that the sign of the Delta A_k, which is essentially what we discuss as the "large-get-stronger, small-get-weaker wave response to warming", would not change when weighting by some density measure. For these reasons we refrain from incorporating your suggestion into this analysis, even though we fully acknowledge its relevance for related analyses of wave energy or EKE.

L148 "we first compute": The order doesn't really matter here, but ok.

We removed "first".

L152-153: Is this supposed to refer to A_k rather than Phi_hov,k? Otherwise, I don't understand what the condition means in terms of a phase. I it is about the amplitude, you could also move this paragraph to the previous subsection.

Yes, thank you very much for spotting this typo – of course we mean A_k!

L165: Remove "(southern)" as you mention SH blocks at the end of the paragraph

Ok, done.

L166: Not sure if the PVU explanation is needed, but if you want to include it I would rather move it to the end of the sentence or so.

PVU is an abbreviation and we thus want to define it where we first use it. We moved it to a footnote.

Eq. 4+5 and L228-229: You could consider to introduce separate parameters for NH and SH (like phi_0,N and phi_0,S or so). That would make the equations look a bit more complicated but might make the description of the overall setup a bit easier to understand.

We tried different variants but felt that the current version in combination with the two sentences following the equations remains our favorite.

Fig. 1: Could you motivate your choices of SST distribution? Later you distinguish between East and West Pacific, do you find your profiles to match these regions?

The first profile is motivated by the mean zonal SST of the JJA and leads to the occurrence of a double ITCZ. As an alternative, we want to create a profile that leads to a single ITCZ by increasing the equatorial SST gradient. They are not motivated by the SST profiles in the East and West Pacific and do not match these better than the zonal mean JJA SST profiles.

L280: The reference (Fig. 5a) seems to refer more to the previous sentence.

The reference was removed.

L290: You discuss Fig. 5 before Fig. 4, maybe swap them or change the discussion?

Indeed, we swapped them.

L338: *isentropic slope (to be more precise)

Thank you, corrected.

Fig. 7: What exactly does the "weighting" refer to? Also, please add sample sizes to the caption.

A weighted histogram is simply a histogram where the counts in each bin have been divided by the total number of counts, with the intention that the resulting histogram can be considered an estimate of an underlying probability density function and thus allows for comparisons across histograms based on different total counts. We now replaced the "weighted" with "normalised", as we feel that "normalised histogram" is a term that is sufficiently standard so that it can be used without further explanations. The sample size varies for each panel, but the range of sample sizes is now given in the caption of Fig. 7.

---

## Author Comment (AC2)

This study explores the variations in atmospheric circulation changes in winter and summer in a warming climate using an idealized aquaplanet set-up, in which the underlying prescribed SST distribution creates a warmer summer hemisphere and colder winter hemisphere. The study focuses on diagnosing changes in tropospheric wave activity, and to a lesser extent blocking and storm track metrics.

Overall, I find the authors' aquaplanet set-up to be an intriguing and effective tool to understand the complicated dynamical changes that can occur in a warming climate. It's great to see that the authors can reproduce many of the same changes that occur in more comprehensive GCMs in an idealized model, suggesting that future analysis of these runs could help to better disentangle the underlying mechanisms.

I see two weaknesses of the study as presently written. First, a detailed discussion of the mean circulation changes with warming seems warranted and is notably absent. Understanding how well the model captures seasonal changes in the mean circulation with warming is important before examining highly derived circulation metrics for extreme weather (waviness, blocking, etc.). Second, the writing is quite sloppy throughout and needs to be improved (see large list of typos below). I would encourage the authors to do a thorough proofreading of their analyses and text before re-submitting to eliminate any careless errors in the final manuscript.

We are grateful to the reviewer for taking the time to carefully review our manuscript and provide constructive feedback. The suggestion to add a paragraph on mean circulation change is appreciated and we have incorporated this into the revised version of the manuscript. We have also taken care to remove all typographical errors and improve the flow of the manuscript to enhance its clarity. A detailed response to the reviewer's comments is provided below.

**Major Revision:**

While the authors' focus here is on understanding changes in extreme weather patterns with warming in summer and winter, it is important for readers to understand how well this aquaplanet model reproduces the mean circulation changes with warming. Some discussion of how the jet stream strength and position changes with warming (and a comparison with results from comprehensive GCMs) seems warranted. For example, if the model doesn't produce a realistic poleward jet/storm track shift with warming, then why should we trust the results of more highly derived circulation metrics? (From Fig. 4, it appears that the jets do shift poleward with warming, but this is not discussed.) The authors focus a lot on vertical shifts in the circulation with warming, but addressing meridional shifts is also relevant.

We agree that this is an important issue. In part, the analysis of the mean circulation change has already been discussed in a previous publication, so we did not investigate it in this submission, but we will add one paragraph on the seasonality of the storm track shift in the revised version. In that earlier paper, the setup was a classic wintertime aquaplanet and it reproduced the poleward shift under warming as well as the enhanced poleward propagation of cyclones. The inclusion of zonal asymmetry even reproduced a pattern of change reminiscent of the North Atlantic response to warming. These results make us confident that the new setup also reproduces the shift under warming.

- Schemm, S., Papritz, L., and Rivière, G.: Storm track response to uniform global warming downstream of an idealized sea surface temperature front, Weather Clim. Dynam., 3, 601–623, https://doi.org/10.5194/wcd-3-601-2022, 2022.

Particularly interesting to me is whether this idealized aquaplanet experiment can reproduce the asymmetry in poleward jet shifts between winter and summer hemispheres seen in comprehensive GCMs. In particular, the Southern Hemisphere and North Atlantic jets experience much larger jet shifts under warming in summer versus winter (see Fig. 12 of Barnes and Polvani 2013). If the authors' aquaplanet model can reproduce this result, it could help to disentangle mechanisms responsible for the seasonality of the mean circulation changes to warming as well.

Barnes, E. A., and L. Polvani, 2013: Response of the Midlatitude Jets, and of Their Variability, to Increased Greenhouse Gases in the CMIP5 Models. J. Climate, 26, 7117–7135

We thank the reviewer for this suggestion. Indeed, in our setup, the storm track shifts further poleward in the warmer compared to the colder hemisphere. More precisely, the EKE maximum shifts poleward by 2° in the colder hemisphere and by 4° in the warmer hemisphere, in agreement with the SH and North Atlantic reported in Barnes and Polvani (2013). This is an exciting result because it confirms the results of previous studies based on comprehensive GCMs and shows that zonally symmetric simulations with reduced complexity can reproduce the seasonality in the storm track response. The underlying processes will be further investigated in the future. We have added a paragraph on this aspect to the revised manuscript.

**Minor Revisions:**

Lines 30-31: I disagree with the authors' assessment that many climate change trends, including those associated with Arctic amplification, are largest in summer. Arctic amplification is actually the smallest in summer and largest in winter (see recent review by Previdi et al. 2021, https://iopscience.iop.org/article/10.1088/1748-9326/ac1c29). Furthermore, projected blocking and storm track trends with climate change are also very pronounced in winter. The general focus on the summer hemisphere throughout the paper is not particularly well motivated in my opinion.

The general focus on the summer hemisphere is motivated by the fact that designing a summer-like APE is one of the main novelties of this study. Furthermore, note that several very high-impact studies have highlighted summer circulation changes (e.g., Coumou et al., 2015, 2018; Rousi et al., 2022) and, moreover, in recent years several types of summertime weather and climate extremes have been a particularly strong societal concern (e.g., numerous exceptionally intense heat waves such as the June 2021 heat wave in the Pacific Northwest, concurrent Northern Hemisphere heat waves in the summer of 2018 and the hot drought in summer 2022 in western Europe). It thus seems warranted to specifically introduce an APE setup with a summer hemisphere.

We do acknowledge that Arctic amplification is indeed strongest in winter and accidentally confused it in the introduction. We have now revised the corresponding paragraph of our introduction section to better motivate the focus on summer circulation changes and to clarify this aspect. Thank you for pointing this out.

Line 100, and hereafter: APE – please define acronym

Done.

Line 130: "same height in pressure space" – This wording is confusing. Do you mean the same fixed vertical pressure level?

Yes, we changed that to "same pressure level".

Lines 148, 152: Wavenumbers 4-8 or 5-8? Please be consistent in your methodology, or explain why there is a difference.

Thank you for spotting this inconsistency. 5-8 is correct, and also consistent with Kornhuber et al. (2020). We corrected that accordingly.

Line 269: I don't see the agreement with reanalysis data here. The jets in the aquaplanet simulations look substantially stronger than in the reanalysis.

We agree. In terms of the zonal mean wind, the jet is stronger than in the reanalysis data. We wanted to point out the many other qualitative agreements between APE and reanalysis data, such as the fact that the winter jet is stronger than the summer jet, and that the storm tracks are closer to the equator. We have clarified this statement.

Section 4.4: To be consistent with the previous subsections in Section 4, a discussion of the blocking results from the idealized model in the context of previous results from comprehensive GCMs is notably absent here.

Our understanding is that there is no robust picture of blocking changes under climate warming emerging from comprehensive GCMs. According to Woollings et al. (2018) there is some evidence for a reduction in blocking frequencies under climate warming, but nevertheless blocking trends often differ in sign across models or indentification scheme and there is no strong consensus. Therefore we refrain from a detailed comparison of our results with results from comprehensive GCMs. However, we do now mention in Section 4.4 (Section 4.5 in the revised manuscript) that the decline in projected blocking frequencies derived from some (pressure-level based) blocking indices (e.g., Woollings et al., 2018) may also be affected by the vertical extension of the troposphere under climate warming, which is the main cause for spurious blocking trends in our simulations.

Lines 419-420: How do you determine whether a region is affected by a cyclone. Presumably, some radius of influence from the cyclone center is defined. It would be good to specify this in the methodology.

The scheme computes cyclone mask, i.e., it flags all grid points inside the outermost closed SLP contour with ones and all other grid points with zeros. The cyclone masks indicate whether a grid point is affected by a cyclone or not. Time averaging across all time steps then yields the relative fraction of times steps a grid point is affected by cyclones. We added this information to the method section. Please find below a screenshot of how the cyclone masks look like at one arbitrary time step.

[Figure]

*Figure 1: Cyclone masks at one time step. Grid points shaded in red indicate the presence of a surface cyclone.*

Line 521: From Fig. 11, it doesn't appear that there is much difference in the decreases in cyclone lifetimes between the summer and winter hemispheres.

This is correct, the sentence should be "*Except for very long-lived cyclones in the colder hemisphere cyclone lifetimes decrease in both hemispheres.*"

Figure 3: This is an odd unit for precipitation. Why not use mm/day?

Ok, $kg/m^2$ corresponds to mm and because it is accumulated over six hours we changed to mm/6h.

Typos:

Line 29: weakened

Corrected, thank you.

Lines 41-42: This is not a complete sentence and does not make sense as written. Please rewrite.

Done, thanks.

Line 58: extent

Corrected, thanks

Line 67: North Atlantic jet or storm track? – something is missing here

Corrected, thanks.

Line 77: per se

Corrected, thanks.

Line 80: different methods

Ok, corrected, thanks.

Line 85: cause

Ok, corrected, thanks.

Line 104: investigated?

Yes, thanks, changed.

Line 151: What is arctan2? Perhaps the 2 is a typo.

No, the 2 is not a typo. The arctan2 is the two-argument arctangent, i.e., the function that maps any (x,y) pair onto the angle between the positive x-axis and the ray connecting the origin and the (x,y) pair. It is thus a very convenient way compute the phase of any complex number, which is exactly why we use it here. We have added a very brief explanation after Eq. (2).

Line 159: 10,000

Ok, changed.

Line 165: Delete "(southern)" here, as that information is given on Line 169.

Ok, changed.

Line 187: 35,000 …. 32,000

Ok, changed.

Line 187: schemes

Ok, changed.

Line 232: double

Ok, changed.

Line 234: experiments

Ok, changed.

Lines 241-242: Not a complete sentence

Ok, completed.

Line 287: occur

Ok, changed.

Line 429: simulations

Ok, changed.

Line 447: 7,000

Ok, changed.

Line 454: West (not Central Pacific) to be consistent with Fig. 2

Changed.

Line 473: held constant

Ok, changed.

Line 479: lower stratospheric (100 hPa in Fig. 6 is in lower stratosphere)

Yes, thanks corrected.

Line 483: waviness

Ok, corrected.

Lines 529-530: allow us to study

Ok, corrected

Line 544, Fig. A1b caption: Text states difference is between 330 K and 340 K isentropic levels, but figure shows 340K and 350K isentropes.

Well spottet.

Fig. 2 caption: blue contours (not blue shading)

Ok, corrected

Fig. 2 caption: flatt -> flat

Ok, corrected

Fig. 6 caption, second line: respectively

Ok, corrected

Fig. 9, legend: 30˚S for winter

Yes!

Table 1: Please use commas not apostrophes in writing the numbers greater than 10,000.

Ok, changed.

---

## Author Response (AR2)

We thank this reviewer, the typos have been corrected.

Line 192, typo: grid point is affected
Line 435: As reported previously by
Fig. A2 caption, Line 589: orange
Line 596: editorial board
Figures 2, 5: missing labels for pressure and altitude on y-axis